# Penigrisacids A–D, Four New Sesquiterpenes from the Deep-Sea-Derived *Penicillium griseofulvum*

**DOI:** 10.3390/md17090507

**Published:** 2019-08-29

**Authors:** Cui-Ping Xing, Chun-Lan Xie, Jin-Mei Xia, Qing-Mei Liu, Wei-Xiang Lin, De-Zan Ye, Guang-Ming Liu, Xian-Wen Yang

**Affiliations:** 1Key Laboratory of Marine Biogenetic Resources, South China Sea Bio-Resource Exploitation and Utilization Collaborative Innovation Center, Third Institute of Oceanography, Ministry of Natural Resources, 184 Daxue Road, Xiamen 361005, China; 2College of Food and Biological Engineering, Jimei University, 43 Yindou Road, Xiamen 361021, China

**Keywords:** deep-sea-derived fungus, *Penicillium griseofulvum*, carotanes, sesquiterpenes, anti-food allergy

## Abstract

Four new (penigrisacids A–D, **1**–**4**) and one known (**5**) carotane sesquiterpenoids were isolated from the deep-sea-derived fungus *Penicillium griseofulvum*, along with four known compounds (**6**–**9**). The planar structures and relative configurations of the new compounds were determined by extensive analysis of the NMR and HRESIMS data. The absolute configurations were established by comparison of the experimental and calculated ECD (electronic circular dichroism) spectra or OR (optical rotation) value. Compound **9** exhibited potent anti-food allergic activity with IC_50_ value of 28.7 μM, while **4** showed weak cytotoxicity against ECA-109 tumor cells (IC_50_ = 28.7 μM).

## 1. Introduction

Carotanes (also called daucanes) are bicyclic sesquiterpenes mainly isolated from the plants of the Umbelliferae family and the fungus *Aspergillus terreus* [1,2,3,4]. Some of these compounds showed significant biological activities, for example, aspterric acid was found to inhibit Arabidopsis pollen development at meiosis [5]. More recently, studies showed aspterric acid could be used as a herbicide [6]. In our current investigations on the bioactive secondary metabolites from the deep-sea-derived microorganisms [7,8,9,10], the crude extract of *Penicillium griseofulvum*, a fungus isolated from the Indian Ocean, showed potent in vitro anti-food allergic activity. Accordingly, a systematic isolation was conducted, which led to the isolation of five carotanes (**1**–**5**, Figure 1) and four other compounds (**5**–**9**). Herein, we report the isolation, structure elucidation, as well as anti-allergic and anti-tumor bioactivities of these compounds.

## 2. Results and Discussion

Compound **1** was obtained as a colorless oil. Its molecular formula was established as C_15_H_22_O_4_ on the basis of the sodium adduct ion peak at *m*/*z* 289.1411 [M + Na]^+^ in its HRESIMS spectrum, indicating five degrees of unsaturation. The ^1^H NMR spectrum (Table 1) showed the presence of three methyl singlets (*δ*_H_ 1.18, s, Me-15; 1.82, s, Me-12; 1.63, s, Me-13) and one oxymethine (*δ*_H_ 4.48, d, *J* = 7.5 Hz, H-2). The ^13^C NMR spectrum, in association with the DEPT and HSQC spectra, indicated 15 carbon signals (Table 2), including three methyls (*δ*_C_ 20.1, 21.3, and 23.5, for Me-15, 12, and 13, respectively), five methylenes (*δ*_C_ 27.7, C-5; 28.5, C-4; 33.3, C-9; 40.4, C-1; 41.6, C-8), one oxygenated methine (*δ*_C_ 82.6, C-2), and six non-protonated carbons, including one carboxyl (*δ*_C_ 178.0, C-14), two olefinic (*δ*_C_ 135.6, C-10; *δ*_C_ 131.6, C-11), two oxygenated (*δ*_C_ 74.4, C-3; 93.9, C-6); and one aliphatic (*δ*_C_ 54.0, C-7) carbons. Since one carboxyl and two olefinic carbons accounted for two degrees of unsaturation, **1** was assumed to be a tricyclic molecule. The COSY correlations showed three isolated spin systems from H-2 (*δ*_H_ 4.48) to H_2_-1 (*δ*_H_ 2.01, 1.90), H_2_-4 (*δ*_H_ 2.18, 1.69) to H_2_-5 (*δ*_H_ 2.74, 1.55), and H_2_-8 (*δ*_H_ 1.63, 1.55) through H_2_-9 (*δ*_H_ 2.38) to Me-12 and Me-13. The HMBC spectrum displayed correlations from Me-15 to C-1/C-6/C-7/C-8, H-2 to C-3/C-4/C-6/C-14, and H_2_-5 to C-6/C-7/C-10. Taking together the COSY and HMBC correlations, the planar structure of **1** was established (Figure 2), which was very similar to aspterric acid except that the ether bond was between C-2 and C-6 in **1**, instead of C-2 and C-15 in aspterric acid. In the NOESY spectrum, strong correlations were observed from H-1a (*δ*_H_ 1.90) to Me-15, H-1b (*δ*_H_ 2.01) to H-2, Me-15 to H-5a (*δ*_H_ 1.55), suggesting the configurations of C-2 and C-6 at the epoxy moiety were opposite to that of Me-15. Accordingly, the relative structure of **1** was assigned as 10,11-dehydro-3α-hydroxy-2α,6α-epoxycarotan-14-oic acid.

To determine the absolute configuration, a theoretical calculation on its ECD spectrum was performed. The calculated ECD spectra of 2*S*,3*R*,6*S*,7*R*-**1** (**1a**) and 2*R*,3*S*,6*R*,7*S*-**1** (**1b**) were obtained by time-dependent density functional theory (TD-DFT) at the B3LYP/6-31+G(d,p) level in ACN (acetonitrile). As shown in Figure 3, the calculated ECD spectrum of **1a** fits well with the experimental one. From the above-mentioned evidences, the structure of **1** was then established as (2*S*,3*R*,6*S*,7*R*)-10,11-dehydro-2,6-epoxy-3-hydroxy-14-carotanoic acid, and was named penigrisacid A.

Compound **2** was also obtained as a colorless oil. The molecular formula C_15_H_24_O_4_ was established according to its positive HRESIMS spectrum at *m*/*z* 291.1563 (calcd for C_15_H_24_O_4_Na, 291.1572), indicating four degrees of unsaturation. The ^1^H NMR spectrum exhibited two methyl singlets (*δ*_H_ 0.79, s, Me-12; 1.19, s, Me-15), one oxygenated methylene (*δ*_H_ 3.45, s, H_2_-13), and one olefinic proton (*δ*_H_ 7.07, dt, *J* = 8.6, 3.1 Hz, H-2). The ^13^C NMR spectrum displayed 15 carbons signals which, in combination with DEPT and HSQC spectra, could be categorized as two methyls (*δ*_C_ 19.2, Me-15; 22.9, Me-12), one oxymethylene (*δ*_C_ 69.5, C-13), five *sp*^3^ methylene (*δ*_C_ 24.6, C-9; 27.0, C-5; 28.5, C-4; 42.3, C-1; 42.4, C-8), two methines (*δ*_C_ 50.1, C-10; 55.8, C-6), one protonated *sp*^2^ (*δ*_C_ 142.8, C-2), and four non-protonated carbons, including one carboxyl (*δ*_C_ 172.0, C-14), one olefinic (*δ*_C_ 136.5, C-3) and one oxygenated (*δ*_C_ 76.3, C-11) carbons. Since one carboxyl and two olefinic carbons accounted for two degrees of unsaturation, **2** was deduced to be a bicyclic molecule. The COSY spectrum exhibited correlations from H-2 (*δ*_H_ 7.07) to H_2_-1 (*δ*_H_ 2.41, dd, *J* = 14.9, 9.1 Hz, H-1a; 2.14, br d, *J* = 14.2 Hz, H-1b) and the overlapped proton at *δ*_H_ 1.93; H-5a (*δ*_H_ 1.25, d, *J* = 13.2 Hz) to H-4a (*δ*_H_ 2.99, dd, *J* = 15.8, 5.8 Hz) and H-6 (*δ*_H_ 1.59, t, *J* = 10.5 Hz); H_2_-9 (*δ*_H_ 1.78 m, 1.68 m) to H_2_-8 (*δ*_H_ 1.41, m) and the overlapped proton at *δ*_H_ 1.93. The HMBC spectrum showed correlations from Me-12 to C-10, C-11, C-13, from Me-15 to C-1, C-6, C-7, C-8, from H_2_-4 to C-2, C-3, C-14, and from H_2_-9 to C-6, C-10, C-11. Taking together the molecular formula, the COSY, and HMBC spectra, the planar structure of **2** was then established. The NOESY spectrum showed correlations from Me-15 to H-1a (*δ*_H_ 2.41)/H-5a (*δ*_H_ 1.25)/H-9a (*δ*_H_ 1.78), from H-1b (*δ*_H_ 2.14) to H-6, and from H_2_-13 to H-5b (*δ*_H_ 1.93)/H-6/H-9b (*δ*_H_ 1.68)/H-10 (Figure 4). Therefore, H-6, H-7, and H_2_-13 were on the same face which was opposite to that of Me-15.

To determine the absolute configuration of the stereogenic carbons of **2**, a theoretical calculation of its ECD spectrum in ACN was performed. As shown in Figure 5, the calculated ECD spectrum of (6*S*,7*R*,10*S*,11*S*)-**2** (**2a**) fit well with that of the experimental one. Accordingly, **2** was elucidated as (6*S*,7*R*,10*S*,11*S*)-2,3-dehydro-11,13-dihydroxy-14-carotanoic acid, and was named penigrisacid B.

Compound **3** showed a molecular formula C_15_H_24_O_5_, as established by its positive HRESIMS spectrum at *m*/*z* 307.1523 (calcd for C_15_H_24_O_5_Na, 307.1521), indicating four degrees of unsaturation. The ^13^C NMR spectrum showed 15 carbon signals which, according to DEPT and HSQC spectra, could be classified as two methyls (*δ*_C_ 26.6, C-12; 29.2, C-13), five methylenes (*δ*_C_ 24.8, C-5; 26.8, C-9; 33.6, C-8; 34.3, C-4; and 35.8, C-1), one oxymethylene (*δ*_C_ 74.5, C-15), two methines (*δ*_C_ 51.8, C-10; 52.3, C-6), one oxymethine (*δ*_C_ 82.8, C-2), and four non-protonated carbons including one carboxyl group (*δ*_C_ 175.8), two oxyquaternary carbons (*δ*_C_ 77.6 and 71.5). The general aspect of the ^13^C NMR spectrum of **3** was similar to that of aspterric acid, except that the signals of the olefinic carbons (C-10/C-11) in aspterric acid were replaced by the signals of one methine (*δ*_H_ 1.58, m, H-10; *δ*_C_ 51.8, C-10) and one oxyquaternary carbon (*δ*_C_ 71.5, C-11) in **3**, respectively. This was confirmed by the HMBC correlations from Me-12 and Me-13 to C-10 and C-11. The NOESY spectrum showed a correlation from H-10 (*δ*_H_ 1.58, m) to H-15a (*δ*_H_ 3.64, d, *J* = 8.0 Hz), indicating H-10 is in the *β* position. Further by comparison of the experimental OR ([α]${}_{D}^{25}$ −49.4) and the calculated one for (2*R*,3*R*,6*S*,7*S*,10*R*)-**3** ([α${]}_{D}^{25}$ −52.9), the full structure of **3** was then established as (10*R*)-10,11-dihydro-11-hydroxyaspterric acid, and was named penigrisacid C.

The molecular formula of **4** was assigned as C_15_H_22_O_5_ on the basis of its positive HRESIMS spectrum at *m*/*z* 305.1365 (calcd for C_15_H_22_O_5_Na, 305.1359), indicating five degrees of unsaturation. The ^1^H NMR spectrum exhibited one methyl singlet (*δ*_H_ 1.76, Me-12), one *sp*^2^ oxymethylene (*δ*_H_ 4.62, d, *J* = 8.0 Hz; 3.34, d, *J* = 7.9 Hz, H-15), one olefinic methylene (*δ*_H_ 5.01, s; 4.84, s, H_2_-13), and one oxymethine (*δ*_H_ 4.28, d, *J* = 8.4, H-2). The ^13^C NMR spectrum in association with the DEPT (distortionless enhancement by polarization transfer) and HSQC (heteronuclear single-quantum correlation) spectra indicated 15 carbon signals ascribed to one methyl (*δ*_C_ 19.7, Me-12); five aliphatic (*δ*_C_ 19.8, C-5; 33.3, C-8; 35.3, C-4; 38.9, C-1; 40.0, C-9), one oxygenated (*δ*_C_ 76.2, C-15), and one olefinic (*δ*_C_ 110.3, C-13) methylenes; one aliphatic (*δ*_C_ 58.4, C-6) and one oxygenated (*δ*_C_ 84.0, C-2) methines; one aliphatic (*δ*_C_ 52.7, C-7), two oxygenated (*δ*_C_ 79.6, C-3; 86.0, C-10), one olefinic (*δ*_C_ 150.7, C-11), and one carbonyl (*δ*_C_ 179.9, C-12) quaternary carbons (Table 2). These signals were very similar to those of **3**, except that the hydroxy moiety at the C-11 in **3** was located at the C-10 position in **4**, while the methyl group at the C-11 position was displaced by a terminal double bond. This was evidenced by the COSY correlations from H-2 (*δ*_H_ 4.28) to H_2_-1a (*δ*_H_ 2.20), from H_2_-5a (*δ*_H_ 1.45) to H_2_-4a (*δ*_H_ 2.24) and H-6 (*δ*_H_ 1.72), from H_2_-8a (*δ*_H_ 1.84) to H_2_-9a (*δ*_H_ 2.10), and from Me-12 (*δ*_H_ 1.76) to H_2_-13 (*δ*_H_ 5.01, 4.84); as well as the HMBC cross peaks of Me-12 (*δ*_H_ 1.76 s) to C-10 (*δ*_C_ 86.0 s), C-11 (*δ*_C_ 150.7 s), and C-13 (*δ*_C_ 110.3 t). Since H-13a (*δ*_H_ 5.01 s) was correlated to H-6 (*δ*_H_ 1.72 m) in the NOESY spectrum, the hydroxy group at the C-10 position was deduced to be *β*-orientation. Therefore, compound **4** was established as 10,11-dihydro-10β-hydroxy-11,13-dehydroaspterric acid, and was named penigrisacid D. All detailed data of theoretical calculations, NMR, and HRESIMS spectra of **1**–**4** can be found in the Appendix A.

By comparison of the NMR and MS data with those in references, the structures of five known compounds were identified as aspterric acid (**5**) [11], aspermytin A (**6**) [12], 1-propanone,3-hydroxy-1-(1,2,4a,5,6,7,8,8a-octahydro-2,5-dihydroxy-1,2,6-trimethyl-1-naphthalenyl) (**7**) [13], craterellone D (**8**) [14], and 3,4-dihydro-3,4,8-trihydroxy-1(2H)-naphthalenone (**9**) [15].

All the nine isolates were tested for anti-food allergic activity. Compound **9** exhibited potent effect with an IC_50_ value of 28.7 μM, compared to 91.6 μM of the positive control, loratadine. While compound **6** showed moderate effect with an IC_50_ value of 97.3 μM. Therefore, **9** could be a promising lead compound for anti-food allergic agent. Moreover, **1**–**9** were also subjected to cytotoxicity assay against five different cancer cells, i.e., BIU-87, Bel-7402, ECA-109, Hela-S3, and PANC-1. However, only **4** showed weak effect on ECA-109 tumor cells with IC_50_ value of 28.7 μM.

## 3. Materials and Methods

### 3.1. General Experimental Procedures

Optical rotations were obtained from an MCP 100 polarimeter (Anton Paar Trading Co. Ltd., Shanghai, China). HRESIMS spectra were conducted on a Xevo G2 Q-TOF mass spectrometer (Waters Corporation, Milford, MA, USA). NMR spectra were recorded on a Bruker 400 MHz spectrometer (Bruker, Fällanden, Switzerland). UV data were recorded from a UV-8000 UV/Vis spectrophotometer (Shanghai Metash instrument Co., Ltd., Shanghai, China). ECD spectra were measured with a Chirascan spectropolarimeter (Applied Photophysic, Beverly, MA, USA). Materials for column chromatography involved silica gel, ODS (octadecylsilyl), and Sephadex LH-20.

### 3.2. Fermentation and Extraction

The fungus was isolated from the deep-sea-sediment (−1420 m) of the Indian Ocean in 2005 by Prof. De-Zan Ye of the Third Institute of Oceanography. It was purchased from the Marine Culture Collection of China with the accession number of MCCC 3A00225. The fungus was cultured on a PDA plate at 25 °C for 3 days. The fresh mycelia and spores were inoculated to 10 × 250 mL Erlenmeyer flasks containing 120 mL of the ISP medium **2** (1.0 L contains 4.0 g yeast extract, 10.0 g malt extract, 4.0 g dextrose, and 20.0 g agar), which were incubated in a 180 rpm rotary shaker at 28 °C for 5 days. Then the spore cultures were used to inoculate 100 × 1 L Erlenmeyer flasks containing corn medium (100 g corn and 120 mL tap water for each flask) to perform the large-scale fermentation. After 62 days, the fermentation broth was extracted with EtOAc three times to provide a crude extract (55.4 g).

### 3.3. Isolation and Purification

The extract was fractionated into six fractions (Fr.1−Fr.6) by column chromatography (CC) on silica gel using a gradient CH_2_Cl_2_-MeOH (0→100%). Fr.4 (4.9 g) was subjected to CC (26 mm × 310 mm) over ODS using H_2_O-MeOH (10→100%) and Sephadex LH-20 (MeOH), followed by purification using preparative TLC (CH_2_Cl_2_-acetone, 2:1) to provide **1** (7.8 mg) and **6** (25.9 mg). Fr.5 (40.0 g) was chromatographed over ODS (H_2_O-MeOH, 5→80%, 49 mm × 460 mm) to get 15 subfractions (Fr.5.1–Fr.5.15), which were further chromatographed on a Sephadex LH-20 (MeOH). Compounds **8** (4.5 mg) and **9** (9.0 mg) were obtained from fractions Fr.5.3 and Fr.5.2 by prep. TLC using EtOAc-MeOH (20:1) and CH_2_Cl_2_-MeOH (10:1), respectively. Fraction Fr.5.4 was subjected to CC over Sephadex LH-20 (CH_2_Cl_2_-MeOH, 1:1), subsequent purification by prep. TLC (PE-EtOAc, 1:2) provided **3** (29.5 mg) and **7** (19.5 mg). Fr.5.5 was subjected to HPLC (MeOH-H_2_O, 20→40%), followed by prep. TLC (CH_2_Cl_2_-MeOH, 10:1) to give **4** (21.4 mg). By prep. TLC using CH_2_Cl_2_-MeOH (10:1), **2** (6.0 mg) and **5** (9.3 mg) were obtained from fractions Fr.5.11 and Fr.5.10, respectively.

Penigrisacid A (**1**): colorless oil; [*α*${]}_{D}^{25}$ +66.2 (c 0.14, EtOH); UV (MeOH) λ_max_ (log ε) 211 (3.29) nm; ECD (ACN) Δ*ε*_209_ +0.32; ^1^H and ^13^C NMR data, see Table 1; Table 2; HRESIMS *m*/*z* 289.1411 [M + Na]^+^ (calcd. for C_15_H_22_O_4_Na, 289.1410).

Penigrisacid B (**2**): colorless oil; [*α*${]}_{D}^{25}$ −29.4 (c 0.11, EtOH); UV (MeOH) λ_max_ (log ε) 209 (3.94) nm; ECD (ACN) Δ*ε*_221_ −0.24; ^1^H and ^13^C NMR data, see Table 1 and Table 2; HRESIMS *m*/*z* 291.1563 [M + Na]^+^ (calcd. for C_15_H_24_O_4_Na, 291.1572).

Penigrisacid C (**3**): colorless oil; [*α*${]}_{D}^{25}$ −49.4 (c 0.16, EtOH); UV (MeOH) λ_max_ (log ε) 201 (2.98) nm; ECD (ACN) Δ*ε*_207_ −0.68; ^1^H and ^13^C NMR data, see Table 1 and Table 2; HRESIMS *m*/*z* 307.1523 [M+Na]^+^ (calcd. for C_15_H_24_O_5_Na, 307.1521).

Penigrisacid D (**4**): colorless oil; [*α*${]}_{D}^{25}$ −30.9 (c 0.41, EtOH); UV (MeOH) λ_max_ (log ε) 209 (3.51) nm; ECD (ACN) Δ*ε*_198_ +0.44; ^1^H and ^13^C NMR data, see Table 1 and Table 2; HRESIMS *m*/*z* 305.1365 [M + Na]^+^ (calcd. for C_15_H_22_O_5_Na, 305.1359).

### 3.4. Theoretical Calculations

As reported previously [7], the preliminary conformational analyses were carried out using RDKit Toolkit [16] by Genetic algorithm at MMFF94 force field. Subsequently, the dominating conformers were re-optimized using density functional theory (DFT) at the B3LYP/6-31+G(d) level. Further calculation in the same level with PCM were conducted for ECD in ACN and OR in MeOH. The ECD spectra of different conformers were simulated by the overlapping Gaussian function [17]. The final spectrum was averaged according to the Boltzmann distribution theory and their relative Gibbs free energy (ΔG).

### 3.5. Anti-Allergic Experiment

As reported previously [18], anti-allergic bioassay was conducted on RBL-2H3 cells. In brief, RBL-2H3 cells were seeded into 96-well cell culture plates to incubate with dinitrophenol specific IgE overnight. IgE-sensitized RBL-2H3 cells were pre-treated with tested compounds for 1 h and stimulated with dinitrophenyl-bovine serum albumin. Phosphate-buffered saline (PBS) buffer and loratadine were used as negative and positive controls, respectively. The bioactivity was quantified by measuring the fluorescence intensity of the hydrolyzed substrate in a fluorometer.

### 3.6. Cytotoxicity Assay

The in vitro antiproliferative assay was performed using MTT method according to the previously reported protocol [19]. Five different cancer cell lines (BIU-87, Bel-7402, ECA-109, Hela-S3, and PANC-1) were seeded into 96-well cell culture plates. After 24 h, different concentrations of tested compounds were added, and the incubation was continued for another 48 h. Then 20 µL MTT solution were added and cell viability was evaluated by measuring the absorbance at 570 nm.

## 4. Conclusions

From the deep-sea-derived fungus *Penicillium griseofulvum*, four new (penigrisacids A–D) and five known compounds (aspterric acid, aspermytin A, 1-propanone,3-hydroxy-1-(1,2,4a,5,6,7,8,8a-octahydro-2,5-dihydroxy-1,2,6-trimethyl-1-naphthalenyl), craterellone D, 3,4-dihydro-3,4,8-trihydroxy-1(2H)-naphthalenone) were obtained. 3,4-Dihydro-3,4,8-trihydroxy-1(2H)-naphthalenone, showed potent anti-food allergic activity (IC_50_ = 28.7 μM), whereas penigrisacid D showed weak cytotoxicity against esophageal cancer cell line ECA-109 tumor cells (IC_50_ = 28.7 μM).

## Figures and Tables

**Figure 1 marinedrugs-17-00507-f001:**
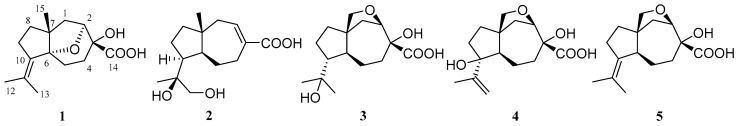
Chemical structures of **1**–**5**, isolated from *Penicillium griseofulvum*.

**Figure 2 marinedrugs-17-00507-f002:**
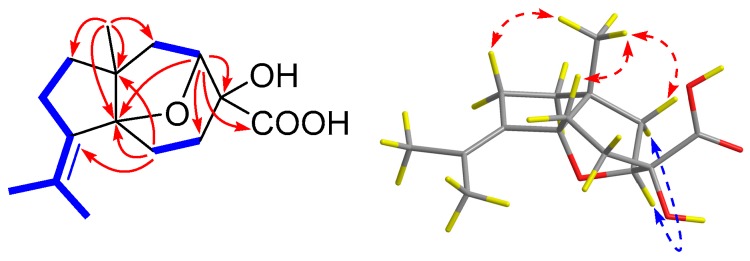
Key COSY (

), HMBC (
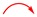
), and NOESY (
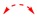
) correlations of **1**.

**Figure 3 marinedrugs-17-00507-f003:**
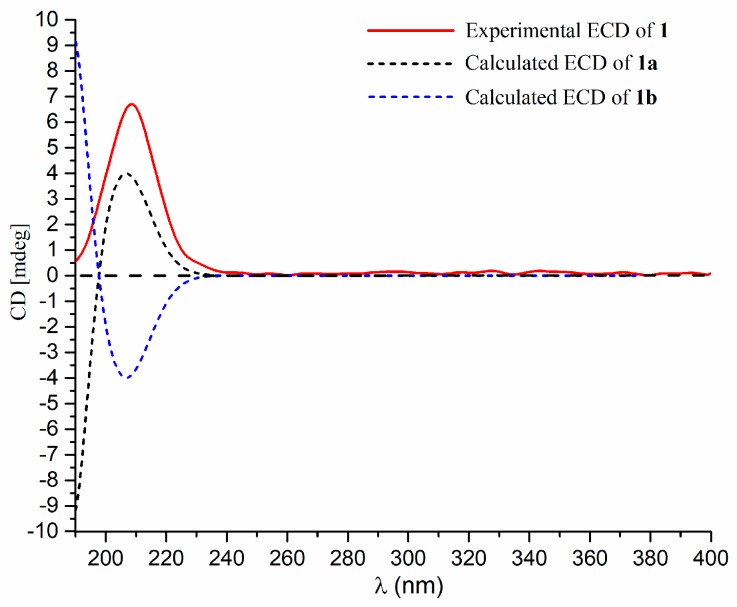
Experimental and calculated ECD spectra of **1**.

**Figure 4 marinedrugs-17-00507-f004:**
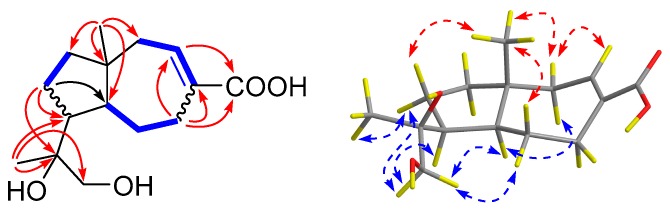
Key COSY (

), HMBC (
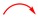
), and NOESY (
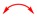
) correlations of **2**.

**Figure 5 marinedrugs-17-00507-f005:**
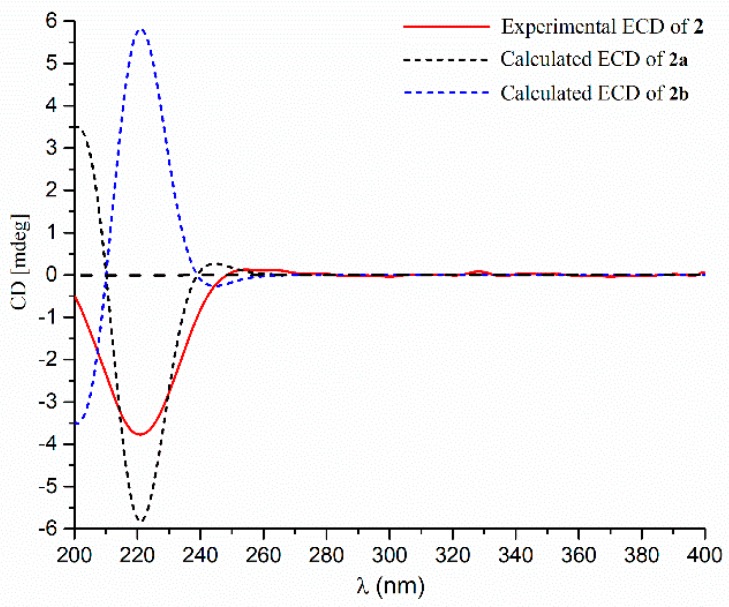
Experimental and calculated ECD spectra of **2**.

**Table 1 marinedrugs-17-00507-t001:** ^1^H (400 MHz) NMR spectroscopic data of **1**−**5** (*δ* in ppm, *J* in Hz within parentheses).

No.	1 ^a^	2 ^a^	3 ^b^	4 ^a^	5 ^a^
1	2.01, dd (13.2, 7.6)	2.41, dd (14.9, 9.1)	2.00, dd (12.7, 8.5)	2.20, dd (13.0, 8.5)	2.40, overlap
	1.90, d (13.2)	2.14, br d (14.2)	1.85, d (12.7)	2.00, d (13.0)	1.48, overlap
2	4.48, d (7.5)	7.07, dt (8.6, 3.1)	4.13, d (8.4)	4.28, d (8.4)	4.31, d (8.4)
4	2.18, dd (13.5, 6.4)	2.99, dd (15.8, 5.8)	2.18, dd (13.8, 7.6)	2.42, dd (13.7, 7.5)	2.28, m
	1.69, m	1.93, overlap	1.20, m	1.33, m	2.03, overlap
5	2.74, dt (12.6, 6.2)	1.93, overlap	1.93, dd (11.0, 8.0)	1.77, m	2.16, dd (13.1, 8.4)
	1.55, overlap	1.25, d (13.2)	1.43, overlap	1.45, dd (13.6, 7.6)	2.03, overlap
6	-	1.59, t (10.5)	1.42, overlap	1.72, m	2.32, m
8	1.63, m	1.41, m	1.56, m	1.84, m	2.40, overlap
	1.55, overlap		1.26, dd (11.1, 7.6)	1.65, m	1.67, m
9	2.38, m	1.78, m	1.41, overlap	2.10, m	1.69, m
		1.68, m		1.61, m	1.48, overlap
10	-	1.93, overlap	1.58, m	-	-
12	1.82, br s	1.19, s	1.03, s	1.76, s	1.73, br s
13	1.63, br s	3.45, br s	1.05, s	5.01, s	1.60, br s
				4.84, s	
15	1.18, s	0.79, s	3.64, d (8.0)	4.62, d (8.0)	3.70, d (8.0)
			3.18, d (8.1)	3.34, d (7.9)	3.35, d (8.7)

^a^ Recorded in CD_3_OD. ^b^ Recorded in DMSO-*d*_6_.

**Table 2 marinedrugs-17-00507-t002:** ^13^C (100 MHz) NMR spectroscopic data of **1**−**5**.

No.	1 ^a^	2 ^a^	3 ^b^	4 ^a^	5 ^a^
1	40.4 CH_2_	42.3 CH_2_	35.8 CH_2_	38.9 CH_2_	35.2 CH_2_
2	82.6 CH	142.8 CH	82.8 CH	84.0 CH	84.9 CH
3	74.4 C	136.5 C	77.6 C	79.6 C	79.5 C
4	28.5 CH_2_	28.5 CH_2_	34.3 CH_2_	35.3 CH_2_	33.0 CH_2_
5	27.7 CH_2_	27.0 CH_2_	24.8 CH_2_	19.8 CH_2_	36.8 CH_2_
6	93.9 C	55.8 CH	52.3 CH	58.4 CH	56.7 CH
7	54.0 C	43.5 C	52.7 C	52.7 C	54.2 C
8	41.6 CH_2_	42.4 CH_2_	33.6 CH_2_	33.3 CH_2_	24.9 CH_2_
9	33.3 CH_2_	24.6 CH_2_	26.8 CH_2_	40.0 CH_2_	35.1 CH_2_
10	135.6 C	50.1 CH	51.8 CH	86.0 C	136.7 C
11	131.6 C	76.3 C	71.5 C	150.7 C	125.2 C
12	21.3 CH_3_	22.9 CH_3_	26.6 CH_3_	19.7 CH_3_	21.0 CH_3_
13	23.5 CH_3_	69.5 CH_2_	29.2 CH_3_	110.3 CH_2_	23.4 CH_3_
14	178.0 C	172.0 C	175.8 C	177.9 C	177.9 C
15	20.1 CH_3_	19.2 CH_3_	74.5 CH_2_	76.2 CH_2_	76.5 CH_2_

^a^ Recorded in CD_3_OD. ^b^ Recorded in DMSO-*d*_6_.

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
