# Peer review of "Penigrisacids A–D, Four New Sesquiterpenes from the Deep-Sea-Derived Penicillium griseofulvum"

_marinedrugs, 2019, doi:10.3390/md17090507_

Round 1

Reviewer 1 Report

It is necessary to provide the CD spectra of compounds 3-5 in Supplementary Material. In the text of the article, it is necessary to give a comparative analysis of the CD spectra of the new compounds 3 and 4 with the CD spectrum of the known aspterric acid (5). The absolute configuration 5 was previously established based on the data of X-ray diffraction analysis, however, in the literature there is no information about CD spectrum of this compound. The spectral data of compounds 1 and 3 are compared in the manuscript with those for aspterric acid. It is necessary to give a link to table 1. Page 2, line 54: it's necessary to delete <epoxide>. An epoxide is a cyclic ether with a three-atom ring. Line 58: accordingly, it's necessary to delete <-epoxy>. Page 5, line 125: it's necessary to change <the methyl group at the C-13 position> to < the methyl group at the C-11 position>. Page 5, line 130,131: it's necessary to change <β-configuration> to<β-orientation> and, accordingly, the structure of compound 4 was established as 10,11-dihydro-10β-hydroxy-…

Page 5. 3.2 Fermentation and Extraction.

Please provide the composition of the corn medium. Please provide the characteristics of the ODC. What is the composition of the system CH2CI2-MeOH was used to obtain Fr.4 and Fr.5? Line 168. The sentence <Fraction Fr.5.4 was further CC on a Sephadex LH-20 > is confusing and need to be rewritten.

Author Response

Response to Reviewer 1#

It is necessary to provide the CD spectra of compounds 35 in Supplementary Material.
Response: As required, the CD spectra of compounds 35 were provided in the Supplementary Material. In the text of the article, it is necessary to give a comparative analysis of the CD spectra of the new compounds 3 and 4 with the CD spectrum of the known aspterric acid (5). The absolute configuration 5 was previously established based on the data of X-ray diffraction analysis, however, in the literature there is no information about CD spectrum of this compound. The spectral data of compounds 1 and 3 are compared in the manuscript with those for aspterric acid. It is necessary to give a link to table 1.
Response: As you can see that compounds 3, 4, and 5 had different ring systems, it is unreasonable to give a comparative analysis of their spectra. Moreover, the authors re-organized table 1 and gave a link in the revised manuscript. Page 2, line 54: it's necessary to delete <epoxide>. An epoxide is a cyclic ether with a three-atom ring.
Response: Thank you for the suggestion. In case of any misunderstanding, the “epoxide ring” was altered to “ether bond” in the revised manuscript. Line 58: accordingly, it's necessary to delete <-epoxy>.
Response: Thank you for the suggestion. Generally, “epoxy” means a cyclic ether. Page 5, line 125: it's necessary to change <the methyl group at the C-13 position> to < the methyl group at the C-11 position>.
Response: Corrected. Page 5, line 130,131: it's necessary to change <β-configuration> to<β-orientation> and, accordingly, the structure of compound 4 was established as 10,11-dihydro-10β-hydroxy-…
Response: Corrected. Page 5. 3.2 Fermentation and Extraction.
Please provide the composition of the corn medium. Please provide the characteristics of the ODS. What is the composition of the system CH2Cl2-MeOH was used to obtain Fr.4 and Fr.5? Line 168. The sentence <Fraction Fr.5.4 was further CC on a Sephadex LH-20 > is confusing and need to be rewritten.
Response: Corrected. The composition of the corn medium was provided. And the sentence was rewritten in the revised manuscript.

Reviewer 2 Report

1)Mass spectra and IR are required for compounds 1-4 and 2D NMR spectra of 1−4 (link not viable).

2)Table 1 is very disorganized.

3) There should be more discussion about anti-allergic activity.

4)Conclusion must be rewritten and must highlight the relevance of the research and the scientific novelty, it is not a summary of the work done.

Author Response

Response to Reviewer 2#

Mass spectra and IR are required for compounds 14 and 2D NMR spectra of 14 (link not viable).
Response: As required, the MS and 2D NMR spectra of 14 were provided in the revised Supplementary Data. Table 1 is very disorganized.
Response: Accordingly, Table 1 was modified in the revised manuscript. There should be more discussion about anti-allergic activity.
Response: Thank you for your nice suggestion. However, with the limited information on the structures and activity, it is difficult to have a useful discussion. Conclusion must be rewritten and must highlight the relevance of the research and the scientific novelty; it is not a summary of the work done.
Response: Thank you very much for the constructive suggestions. The main purpose of the manuscript is to report four new carotane sesquiterpenoids along with their anti-allergic activity.

Round 2

Reviewer 2 Report

Dear authors is necessary to include infrared (IR) signal in the text for each compound 1-4 . Also in supplementary material it would be good to place IR spectra of the analyzed compounds.

Author Response

Dear Reviewer,

Thank you very much for your nice suggestion. Unfortunately, after the cytotoxic and anti-food allergic bioassays, no amount was left for compounds 1 and 2, which were previously obtained with 7.8 mg and 6.0 mg, respectively. Normally, IR data should be measured for all new compounds to identify the functional groups. However, as to the four new compounds reported in our manuscript, the functional groups including the olefinic bond, carboxyl, epoxy, and hydroxy moieties could be easily recognized on the basis of their 1D and 2D NMR data, and further confirmation could be found using HRESIMS. Therefore, the authors thought it might be OK without the IR data for compounds 14, which I hope you could understand. Thank you very much.

Best regards,

Xianwen YANG
Professor of Natural Product Chemistry

Key Laboratory of Marine Genetic Resources
Third Institute of Oceanography
State Oceanic Administration
184 Daxue Road
Xiamen 361005, China
Email: yangxianwen@tio.org.cn